# Acute Effects of a Single Bout of Walking on Affective Responses in Patients with Major Depressive Disorder

**DOI:** 10.3390/ijerph18041524

**Published:** 2021-02-05

**Authors:** Martin Niedermeier, Larissa Ledochowski, Hartmann Leitner, Helmut Zingerle, Martin Kopp

**Affiliations:** 1Department of Sport Science, University of Innsbruck, 6020 Innsbruck, Austria; info@praxis-psychologie.com (L.L.); hartmann.leitner@hotmail.com (H.L.); Martin.Kopp@uibk.ac.at (M.K.); 2Center for Mental Health Rehabilitation Bad Bachgart, 39030 Rodeneck, Italy; zingerle_helmut@yahoo.de

**Keywords:** Circumplex Model, depression, exercise, physical activity, well-being

## Abstract

Exercise programs are considered an effective (add-on) treatment option for depressive disorders. However, little is known about the acute effects of exercise on affective responses in in-patient settings. Therefore, the aim of the present study was to compare the effects of a single 30-min bout of walking on affective responses to a passive control condition in patients with major depressive disorder during treatment in a mental health center. In total, 23 in-patients were exposed to two conditions (duration: 30 min) using a within-subject design: an exercise (light–moderate walking outdoors) and a passive control condition (sitting and reading). Affective responses were assessed based on the Circumplex Model in four phases: pre, during, post, and two hours following the conditions. The main results include a significantly larger pre–post increase in energetic arousal in the exercise condition compared to the control condition, *p* = 0.012, η_p_² = 0.25, but no significantly different pre–follow-up change between conditions, *p* = 0.093, η_p_² = 0.12. Negatively valenced affective responses showed significantly stronger pre–post decreases after the exercise condition compared to the passive control condition, *p* < 0.036, η_p_² > 0.18. Positively valenced affective response activity showed a pre–post increase in the exercise condition and a pre–post decrease in the passive control condition, *p* = 0.017, η_p_² = 0.23. The higher-activated, positively valenced immediate response of light- to moderate-intensity walking may serve as an acute emotion regulation in patients with major depressive disorder and provide a favorable state for adherence to exercise programs.

## 1. Introduction

According to the Global Burden of Diseases, Injuries, and Risk Factors Study 2010 [1], mental and substance use disorders show the highest burden of disability worldwide. In numbers, mental and substance use disorders account for more than 180 million disability-adjusted life years, which is equal to 7.4% of all disability-adjusted life years globally [1]. With an estimated 322 million cases worldwide, depressive disorders are the most prevalent disorders among mental and substance use disorders and are considered the single largest contributor to the global burden of disability [2]. Therefore, not only an effective prevention of depressive disorders but also effective treatment options against depressive disorders are of utmost importance. As outlined by the National Institute of Clinical Excellence, two of the recommended and most widely used treatment options are (a) medication and (b) psychological interventions [3]. In addition, exercise programs are recommended in the treatment of depressive disorders [3]. On the one hand, exercise programs can reduce depressive symptoms [4,5]. On the other hand, exercise programs are often used as an add-on treatment to buffer risks connected to the standard treatment, e.g., reducing adverse side effects of medication [6] or bridging the lag time between starting the standard treatment and reductions in depressive symptoms [4]. Although the long-term effects of exercise programs are of essential importance, recent research also focuses on acute effects of single bouts of exercise. Acute effects of exercise have been studied less intensively compared to long-term effects of exercise programs, especially in in-patients suffering from depressive disorders [7,8,9,10,11]. However, knowledge on the acute effects of single bouts of exercise on psychological outcomes in in-patients is important due to a variety of reasons. Firstly, single bouts of exercise may improve well-being and vigor in people suffering from major depressive disorder [12], thereby serving as a form of emotion regulation, which is considered an important component in the treatment of patients with mental health disorders [4]. Secondly, acute exercise effects may lead to a better clinical response in standardized clinical anti-depressive medication by helping to bridge the time until pharmacological effects are reached [7,13]. Thirdly, acute exercise effects may lead to an overall better clinical outcome by improving treatment responses to medication as well as to cognitive-behavioral therapies [13]. Fourthly, it might be important for the treatment outcome to consider the patients’ preferences for the treatment options [14], and some people may prefer alternative treatment such as exercise [4,15]. Preferences might be more unconsciously assessed by affective responses during and after the activity. Fifthly, and connected with the previous point, adherence to the treatment, which is a major issue in patients with depressive disorder [3], is larger when affective responses are positively affected by the treatment [12]. In this context, it is valuable to know if exercise can improve acute psychological outcomes in in-patients.

A key component in acute psychological outcomes of exercise is affective responses. Affective responses in the context of exercise are considered as the product of the continuous interplay between physiological signals of the body and cognitions related to self-efficacy, worries, goals, or expectations [16]. Affective responses seem to be a major factor in human decision making [17,18]. It has been shown that the affective components of pleasure and activation may be crucial for bridging the intention–behavior gap at the beginning of exercise engagement [17,19]. The Circumplex Model proposed by Russell [20] is based on the idea that affect is defined by two basic orthogonal, bipolar dimensions (affective valence and perceived activation). The combination of affective valence and perceived activation results in four quadrants, where the upper right quadrant (a positively valenced high-activated state) is the ideal quadrant for exercising [21]. The basic dimensions can be rotated by 45° in the Circumplex Model, resulting in two different axes covering the combination of affective valence and perceived activation and pointing directly in the ideal upper right quadrant [22]. In addition to the basic unrotated and rotated dimensions, categorical affective states are conceived as combinations of these two basic ingredients in different degrees [23]. According to the categorical approach, various distinct states can be specified, e.g., tension, anger, vigor, fatigue, or confusion [24].

Previous research has supported the Circumplex Model of affect in clinical samples [25,26]. There is limited evidence in patients with depressive disorder that single exercise bouts can have positive effects on affective responses [10,11,12]. Exercise intensity was in the moderate range in two of the studies [10,12], which might be perceived as challenging in patients who are less used to exercise. In the other study, different modes of exercise (climbing and swimming) with longer duration (60 min) were studied in adolescent patients [11]. Climbing and swimming are more resource-intense compared to walking. Furthermore, different concepts of questionnaires were used in the existing studies, a between-groups design was used [12], or no follow-up measurement was conducted [10]. Therefore, little is known on short (30 min) exercise bouts with lighter intensity using a within-subject study design. Comprehensive information on different concepts of the Circumplex Model (rotated dimensional, unrotated dimensional, and categorical approach) within a single study is missing. Following these considerations, the present study aimed to investigate immediate and potential longer-term (two hours after the bout) effects of a single bout of walking on affective responses compared to a passive control condition in individuals with major depressive disorder in an in-patient setting. Based on the available literature, we hypothesized that a single bout of walking might show more favorable immediate effects on affective responses (i.e., more positively valenced/higher-activated state) compared to a passive control condition. To attain a comprehensive overview on the characteristics of exercise-induced affective responses in the present population, a variety of questionnaires assessing affective responses based on the Circumplex Model were chosen.

## 2. Materials and Methods

### 2.1. Design and Participants

In a within-subject crossover design, all patients completed two conditions at the same time of day on two separate days: an exercise condition and a passive control condition. Starting order was randomized to minimize familiarization effects. All participants read and signed an informed consent form prior to their involvement in the study. The study was approved by the Institutional Review Board of the Department of Sport Science, University of Innsbruck (#071/2013).

All participants were in-patients during psychiatric treatment in a mental health center and were either diagnosed with F.33 (major depressive disorder, recurrent) or F.32 (major depressive disorder, single episode) [27]. Some of the patients suffered from comorbid mental disorders, e.g., F.13 (sedative, hypnotic, or anxiolytic-related disorders), F.41 (other anxiety disorders), F.43 (reaction to severe stress, and adjustment disorders), F.50 (eating disorders), or F.55 (abuse of non-psychoactive substances) [27]. Inclusion criteria were (a) diagnosed depressive disorder and (b) total Beck Depression Inventory II score ≥ 18. Exclusion criteria were (a) acute injury or illness that would inhibit their ability to safely exercise when undergoing study conditions (medically confirmed) and (b) inability to exercise with a moderate intensity, assessed by the Physical Activity Readiness Questionnaire [28]. All patients not meeting the exclusion criteria and who expressed interest in the study were approached during the study period to attain the largest sample size possible.

### 2.2. Procedure

The procedure was identical in both conditions, except for the activity, which contained exercise one time and a passive control the other time, each lasting 30 min. Data were collected at a maximum of eight time points in four phases per each condition (Figure 1): immediately prior to the condition (pre), during condition (5, 10, 15, 20, and 25 min after the start of the condition), immediately after the condition (post), and two hours after the condition (follow-up). Four different sets of self-reported questionnaires based on the Circumplex Model [20] were used for the assessment of affective responses (see Section 2.4). Both a dimensional and categorical approach were chosen [24]. The categorical approach gives information on distinct states of affect (e.g., anger, vigor, and fatigue), while the basic dimensional approach covers the underlying dimensions of affect (i.e., affective valence and perceived activation). The dimensional approach was further differentiated by (a) the basic dimensions affective valence and perceived activation and (b) the rotated dimensions of energetic arousal and tense arousal, which show a rotation of 45° in the Circumplex Model compared to the basic dimensions [29]. Assessment of the basic dimensions can be performed using two single-item scales, which offers the possibility to assess them relatively quickly and to investigate changes over short-term periods (i.e., minutes), while completion of the multi-item scales to assess distinct categories and rotated dimensions of affect is more time-consuming. Therefore, basic dimensional affective responses were assessed at all time points. Rotated dimensional affective responses using a multi-item scale were collected at three time points, i.e., pre, post, and follow up. Categorical affective responses were collected at two time points, i.e., pre and post. At all five time points during exercise/passive control, information on perceived exertion was collected. Heart rate was assessed throughout the conditions. Participants returned to their normal daily routine in the institution after exercise/passive control and returned for the follow-up to complete the questionnaires. In both conditions, the interaction between participant and investigator was kept to a minimum, with standardized instructions provided.

### 2.3. Conditions

The exercise condition contained walking outdoors (on a standardized cross-country course) with a self-selected intensity in the light–moderate range (duration: 30 min). As a guideline for selecting a “brisk” walking pace (as per the current exercise recommendations), participants were told to imagine that they were walking as if they were in a hurry to catch a bus in accordance with previous studies [30,31]. The self-selected intensity in the light–moderate range was selected on previous findings in healthy persons that walking on a self-chosen intensity can cause positive changes in affect [21].

The passive control condition involved sitting passively in quiet place with access to reading materials (usual waiting room reading material—glossy papers) for 30 min.

### 2.4. Measurements Section

#### 2.4.1. Basic Dimensional Affective Responses

Affective valence was assessed by the Feeling Scale (FS) [32]. This single-item rating scale ranges from +5 (“very good”) to −5 (“very bad”), with anchors at zero (“neutral”) and at all odd integers. Perceived activation was assessed by the Felt Arousal Scale (FAS) [33]. This single-item rating scale ranges from 1 (“low arousal”) to 6 (“high arousal”). Both the FS and FAS have been used in previous exercise studies, demonstrating acceptable convergent validity values for the English (*r* = 0.45 to 0.88) and the German versions (*r* = 0.50 to 0.73) [34,35].

#### 2.4.2. Rotated Dimensional Affective Responses

The Activation Deactivation Adjective Check List (AD ACL) [22] is a 20-item measure of two bipolar dimensions, namely energetic arousal and tense arousal. Energetic arousal extends from positive valence/high activation (e.g., energetic, lively) to negative valence/low activation (e.g., tired, drowsy), and tense arousal extends from negative valence/high activation (e.g., tense, jittery) to positive valence/low activation (e.g., calm, at rest). The AD ACL was administered with its standard instructions and its 4-point response scale, which ranges from “definitely feel” to “definitely do not feel”. After recoding reversely coded items, sum scores ranging from 10 (low energetic/tense arousal) to 40 (high energetic/tense arousal) were calculated. Reliability for the subscales energetic arousal and tense arousal were *r* = 0.88 and *r* = 0.91, respectively (German version of the AD ACL) [36]. Internal consistency was between Cronbach’s α = 0.81 and 0.88 [36].

#### 2.4.3. Categorical Affective Responses

The Mood Survey Scale (MSS) [37] assesses eight categorical states (activity, elation, calmness, fatigue, depression, contemplation, anger, and excitement) with a total of 40 items (“At this moment, I feel…”) answered on a 5-point Likert scale. The subscales consisting of the sum score of five items range from 5 (lowest value) to 25 (highest value). Internal consistency values for the subscales range between Cronbach’s α = 0.70 and 0.88 [37]. Convergent validity coefficients for the subscales are between *r* = 0.54 to *r* = 0.87 [37]. The MSS has been recommended for within-subject studies that compare exercise-induced feeling values [38] and is a frequently used instrument for affective responses in German-speaking countries [10,13,39,40].

#### 2.4.4. Physical Exertion

The Rating of Perceived Exertion (RPE) [41,42] was used to assess perceptions of effort during the conditions. The scale ranges from 6 (“extremely light”) to 20 (“extremely hard”). Borg [41] has provided extensive reliability (*r* > 0.90) and validity information (*r* > 0.80) on the RPE. A heart rate monitor (RS800CX, Polar, Finland) was used to collect heart rate data continuously and average heart rate during the conditions was calculated. Subsequently, the intensity of the exercise condition was expressed as percentage of heart rate reserve [43].

### 2.5. Statistical Analyses

Statistical evaluation was performed using SPSS version 26 (IBM, New York, NY, USA). A series of fully repeated measures analyses of variance (rANOVA) were performed to examine the main and interaction effects of condition (2 levels: exercise and passive condition) and time (2, 3, or 8 levels) on affective responses. The levels of the factor time were different between affective responses according to the time points assessed: basic dimensional affective responses (8 levels), rotated dimensional affective responses (3 levels), and categorical affective responses (2 levels). Significant interactions between condition and time were considered as different changes in affective responses between conditions. The interaction analysis was seen as the primary analysis of interest. Pre-planned contrasts using the time point “pre” as the reference category were conducted for the factor time. Whenever the assumption of sphericity according to Mauchly’s test was not met in the rANOVAs (except for categorical affective responses), the Greenhouse–Geisser correction technique was applied. Although a variety of outcomes were tested, we abstained from applying a Bonferroni correction due to multiple outcomes, since we did not want to increase the likelihood of type II errors [44].

Partial eta squared (η_p_²) was calculated as effect size for all rANOVAs. The level of significance was set at *p* < 0.05 (two-tailed). Unless otherwise stated, data are presented as means (SD).

## 3. Results

### 3.1. Preliminary Analysis

All 23 patients (61% female, mean body mass index: 23.9 (5.3) kg/m², age: 48.5 (12.1) years, Beck Depression Inventory II score: 25.3 (5.5)) completed the study. It was the first in-patient treatment for most patients (91%). The majority (87%) received pharmacological medication because of depressive disorder. The rate of current smokers was 44%, and 48% stated that they had experience in exercise programs prior to the study start. Average perceived exertion in the exercise condition translated to “fairly light”: 11.0 (0.8). The average heart rate was 112 (2) bpm during the exercise condition and 76 (4) bpm during the passive control condition. Mean exercise intensity was at 38% of heart rate reserve.

### 3.2. Basic Dimensional Affective Responses

No significant condition by time interaction was found for affective valence (*F*(2.7, 59.0) = 0.80, *p* = 0.487, η_p_² = 0.035) or perceived activation (*F*(3.2, 70.2) = 0.18, *p* = 0.920, η_p_² < 0.01), indicating a similar development over time in both conditions (Figure 2).

A significant main effect of time was found for affective valence (*F*(2.5, 54.7) = 8.18, *p* < 0.001, η_p_² = 0.27) and perceived activation *(F*(2.8, 62.3) = 3.73, *p* = 0.017, η_p_² = 0.15). Affective valence increased over time and perceived activation decreased over time. No significant main effects of condition were found for affective valence and perceived activation (*F*(1, 22) < 0.69, *p* > 0.418, η_p_² < 0.03).

### 3.3. Rotated Dimensional Affective Responses

A significant condition by time interaction emerged for the dimension energetic arousal (*F*(2, 44) = 3.77, *p* = 0.031, η_p_² = 0.15) (Figure 3). Simple contrasts showed a significantly larger pre–post increase in energetic arousal in the exercise condition compared to the control condition (*p* = 0.012, η_p_² = 0.25) but no significantly different pre–follow-up change between conditions (*p* = 0.093, η_p_² = 0.12). No significant condition by time interaction was found for the dimension tense arousal (*F*(2, 44) = 1.09, *p* = 0.345, η_p_² = 0.05).

A significant main effect of time was found both for dimension energetic arousal (*F*(2, 44) = 4.96, *p* = 0.011, η_p_² = 0.18) and tense arousal (*F*(2, 44) = 18.65, *p* < 0.001, η_p_² = 0.46). On average, energetic arousal significantly increased from pre to post (*p* = 0.045, η_p_² = 0.17) and from pre to follow-up (*p* = 0.008, η_p_² = 0.28). Tense arousal significantly decreased from pre to post (*p* < 0.001, η_p_² = 0.46) and from pre to follow-up (*p* < 0.001, η_p_² = 0.61). No significant main effects of condition were found for rotated dimensional affective responses (*F*(1, 22) < 1.48, *p* > 0.237, η_p_² < 0.07).

### 3.4. Categorical Affective Responses

Significant condition by time interactions were found for the categorical affective responses activity, fatigue, anger, and excitement (Table 1). Negatively valenced affective responses fatigue, anger, and excitement showed mean decreases in both conditions, with a stronger decrease after the exercise condition compared to the passive control condition. Positively valenced affective response activity showed a pre–post increase in the exercise condition and a pre–post decrease in the passive control condition.

A main effect of time was found for all categorical affective responses except for activity. Negatively valenced affective responses fatigue, depression, anger, and excitement showed decreases in both conditions. Positively valenced affective responses elation and calmness showed a pre–post increase in both conditions. Neutral affective response contemplation showed a pre–post decrease. There was a main effect of condition for depression, anger, and excitement. Depression, anger, and excitement were rated higher in the exercise condition compared to the passive control condition.

## 4. Discussion

The aim of the present study was to compare immediate and potential longer-term effects of a single bout of walking on affective responses to a passive control condition in patients with major depressive disorder. Using a variety of questionnaires assessing affective responses based on the Circumplex Model showed significantly more positive pre–post changes in energetic arousal and distinct categorical affective responses (activity, fatigue, anger, and excitement) in the exercise condition compared to the passive control condition. This means that a single bout of walking can result in improved affective responses after walking and partly confirms our stated hypothesis. No significant differences between conditions were found for pre–follow-up changes, suggesting reduced likelihood of longer-term effects two hours after the conditions.

### 4.1. Affective Responses to Exercise Bouts in In-Patient Settings

Previous studies in the clinical context have shown that single bouts of moderate-intensity exercise can have positive short-term effects on affective responses in patients with depressive disorder [10,11,12]. Similar decreases in negatively valenced responses were reported after a moderate 30-min exercise bout compared to a quiet rest condition in a study using a between-subjects design [12]. However, vigor and positive well-being were significantly increased in the exercise group only [12], which is similar to the findings of the present study. Mean exercise intensity according to the Ratings of Perceived Exertion was 13.2, translating to “somewhat hard” [12]. A longer exercise bout conducted outdoors (walking for 60 min) was even superior in affective improvements compared to indoor (cycling on an ergometer) and sedentary equivalents for self-reported excitement and activation [10]. The mean exercise intensity according to the Ratings of Perceived Exertion was similar to the previous study (walking: 13.3, cycling: 13.8) [12]. Another mode of exercise (climbing for 60 min) resulted in higher pre–during changes in affective valence compared to swimming or occupational therapy in an adolescent clinical sample [11]. However, similar pre–post changes were found in the three physically active conditions with light exercise intensity (mean Ratings of Perceived Exertion in occupational therapy: 8.3, swimming: 9.6, climbing: 11.9) [11].

The present study extends the existing knowledge of immediate and longer-term effects of a single bout of exercise on affective responses in several aspects and has important implications both for research and for the therapy setting. Firstly, low-intensity exercise can—similarly to more intense exercise—result in positively valenced immediate responses [12]. Therefore, low-intensity exercise might be an adequate option in patients who are less physically active or less used to exercise. Low-intensity exercise may be used more intensively at the start of pharmacological treatment to bridge the phase of the pharmacological action latency [7]. Putting the results in a broader context, exercise programs with self-chosen intensity, associated with positive affective responses, may lead to greater participation in exercise programs among patients with major depressive disorder, as this has been shown for healthy people [45]. Secondly, a comprehensive insight into affective responses was attained with a variety of questionnaires based on the Circumplex Model, which was not present to this extent in the clinical context previously. Thirdly, although a longer-lasting positive change in affect due to walking occurs in terms of emotion regulation [4], more positively valenced affect two hours after treatment is unlikely according to the present results. When longer-term effects were assessed in clinical settings, favorable changes in affect were visible after 30 min of exercise, but not after 60 min or later [12]. The present results support this finding and suggest only transient changes in affect.

Although it has been suggested in earlier literature [46] that light-intensity and short-duration exercise is not enough to result in sufficient changes in affect, the present study is in line with Ekkekakis, Hall, Van Landuyt, and Petruzzello [21], who have shown that even short (10–15 min) walks at a self-chosen intensity (on average, 15–22% of heart rate reserve) can cause changes in affect in healthy persons. The results of the present study show that patients with major depressive disorder experience similar psychological benefits to those reported for healthy participants, i.e., that a short walk outdoors leads to more energy and higher activation [29] as well as reductions in anger, excitement, and fatigue [40,47]. It should be noted that nearly half of the sample did not have experience in exercise programs prior to the study start. Affective responses to physical activity bouts might be dependent on the physical activity level, i.e., a sedentary lifestyle might result in negatively valenced affective responses after a physical activity bout—especially when conducted at higher intensity [48]. Practically speaking, it is important not to overburden patients who are inexperienced with physical activity. Rather than insisting on a specific intensity, a self-selected intensity might be chosen at the start of an exercise program [48].

### 4.2. Critical Reflection of Mixed Findings

The present findings on rotated dimensional and categorical affective responses are consistent. The walking bout showed positive changes in most distinct categorical affective responses, specifically activity, fatigue, and excitement. These categorical states are on the energetic arousal axis in the Circumplex Model [20,37]. However, the findings on the basic (unrotated) affective responses do not perfectly correspond to our other findings and to the hypothesis stated. Given the present finding that patients feel higher energetic arousal after the exercise condition and in accordance with previous literature [10,11,12], a similar effect in affective valence and/or perceived activation was expected which was not present and is hard to explain. Reasons for the discrepancy might be found in the nature of the scales (single- vs. multi-item scale) and in the theoretical concept of the rotated affective responses, e.g., the calmness scale, which theoretically represents pleasant low activation and includes items such as “still” and “quiet”, which others consider as being neutral in terms of valence [49].

Given previous findings in clinical and healthy populations [10,29,40,50], comparable changes in affective valence and perceived activation between exercise and passive control conditions were an unexpected result, which did not confirm our hypothesis. In the exercise condition, we expected to see a stronger increase in affective valence and perceived activation from pre-to-post conditions [40] and pre-to-during conditions [11]. The high activation state prior to both conditions (as visible in Figure 2) might reflect nervousness connected to the experimental situation and might have masked condition-specific changes in affect. For the exercise condition, this statement is supported by higher anger and excitement, which correspond to medium- or high-activation states [37]. Other explanations might be found in an enlarged cognitive capacity for rumination while sitting, which could be related to an increase in activation or in the repeated measurements of basic (unrotated) affective responses (eight times per condition), potentially resulting in a less focused completion of the questionnaires.

Mechanisms behind the changes in affective responses were beyond the scope of the present study. However, future research focusing on physiological measures, such as blood sampling for the determination of serotonin level or salivary cortisol concentrations [51], but also assessments with near-infrared light, positron emission tomography, electroencephalography, and single-photon emission computerized tomography might help to gain more insight on changes in affect. These studies might help to clarify whether these changes are driven by previously theorized decreased brain activity in the prefrontal cortex during activity followed by enhanced prefrontal function after the activity or other mechanisms [52]. In addition, future research on mechanisms should take into account explanations stemming from socio-cognitive variables, including increased distraction during exercise [53] and higher perceived self-competence or social support during and after exercise [54].

### 4.3. Limitations and Strengths

The following limitations need to be considered when interpreting the present findings. Firstly, we did not control for the patients’ activities between leaving our experimental setting and the two hours until the follow-up condition assessment. In the present study, we assumed similar activities and similar affective responses between post and follow-up in both conditions, which may not have been the case. Future research might consider controlling the participants’ activities between finishing the activity and the last measurement point. Secondly, although the sample size is in the range of previously conducted studies in the field [10,12], the sample size (*n* = 23) is relatively small. Future studies should seek larger sample sizes. Thirdly, we cannot rule out a possible selection bias in the form that the patients studied had a certain interest in exercise. This level of interest might not be valid for all patients with major depressive disorder. Assessment of preferences for exercise is recommended for future research. Fourthly, it is possible that participants tried to please the experimenter with their answers. To avoid such examiner-induced effects, it might be useful to reduce the level of interaction between the subject and tester in laboratory conditions with the help of technical aids such as touch screens. On the other hand, this approach might be impaired by reduced ecological validity. Fifthly, although we followed an approach frequently conducted in exercise psychology [24,29], it must be critically discussed whether the large number of completions of the Feeling and Felt Arousal Scales results in a well-reflected filling out of the questionnaires or if the interruption of the activity due to the measurement might affect the results of the study.

The major strength of the present study is the sample of in-patients with major depressive disorder, who are relatively rarely studied. Furthermore, we consider the within-subject design to control for inter-individual variation as a strength. We also want to point out the use of a variety of questionnaires assessing affective responses based on the Circumplex Model and a follow-up assessment two hours after the conditions.

## 5. Conclusions

In conclusion, the present study shows that a single session of light-to-moderate-intensity walking can have positive short-term effects on affect among people with major depressive disorder. Therapists could help patients to use these effects as an option for acute emotion regulation, i.e., to immediately increase energetic arousal and to reduce negatively valenced states such as anger and fatigue. Although longer-term effects of a walking bout on affective responses seem unlikely, a single light-to-moderate-intensity walking bout can eventually serve as a first step towards regularly using exercise to regulate affect, particularly energetic arousal. Once this positive association between exercise and affect is experienced, patients can be encouraged for further participation in exercise programs by reinforcing these benefits.

Future research might consider focusing on the mechanisms behind affective responses. For target group-oriented development of effective exercise programs in the treatment of depressive disorders, an explanation for the experimentally confirmed relationship of exercise and affect is important. Additionally, future studies might explore whether light–moderate-intensity exercise at the beginning of psychiatric treatment improves the therapeutic outcome or not.

## Figures and Tables

**Figure 1 ijerph-18-01524-f001:**
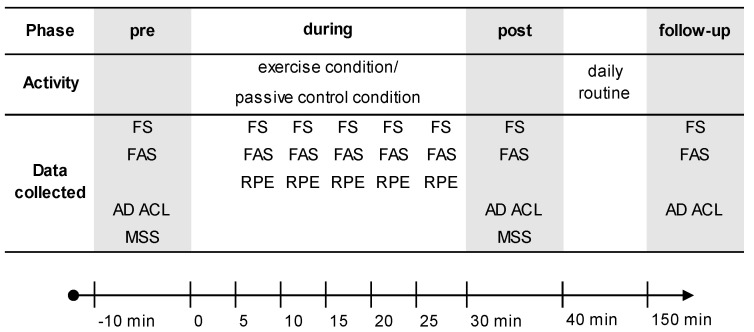
Study procedure including data collection points for each condition. Heart rate was measured throughout the condition. FS: Feeling Scale; FAS: Felt Arousal Scale; RPE: Ratings of Perceived Exertion; AD ACL: Activation Deactivation Adjective Check List; MSS: Mood Survey Scale.

**Figure 2 ijerph-18-01524-f002:**
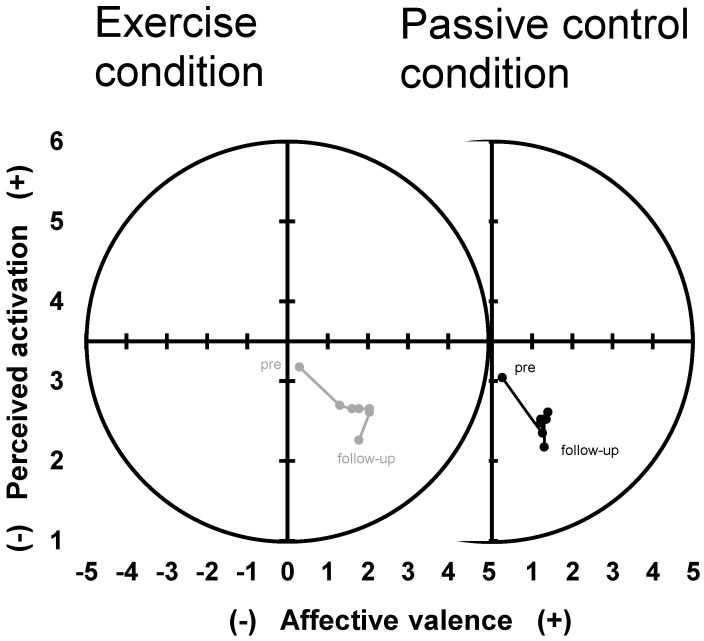
Development of dimensional affective responses affective valence and perceived activation over time in the exercise condition (left panel, grey dots/line) and the passive control condition (right panel, black dots/line). Dots represent mean values averaged for each time point.

**Figure 3 ijerph-18-01524-f003:**
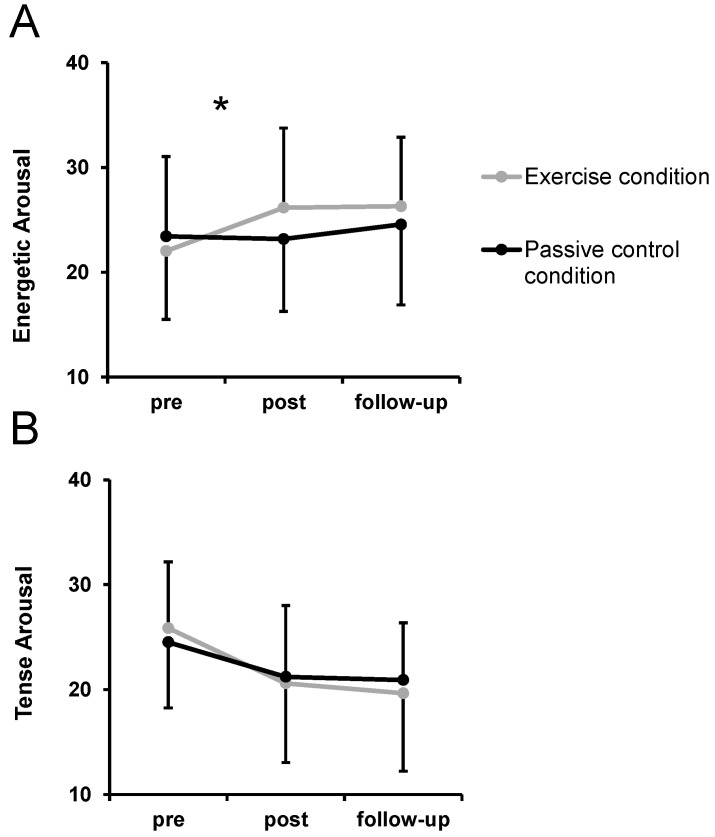
Development of energetic arousal (**A**) and tense arousal (**B**) over time in the exercise condition and the passive control condition. Dots represent mean values and error bars represent standard deviations. * indicates a significant condition by time interaction.

**Table 1 ijerph-18-01524-t001:** Mean (SD) values for categorical affective states by condition and time points including *p*-values and effect sizes.

Categorical Affective States	Exercise Condition				Passive Control Condition				*p*-Value			η²p ^a^		
	Pre		Post		Pre		Post							
	M	(SD)	M	(SD)	M	(SD)	M	(SD)	Condition	Time	Interaction	Condition	Time	Interaction
Activity	11.2	(3.7)	13.1	(3.8)	12.1	(3.9)	11.3	(3.5)	0.271	0.329	**0.017**	0.05	0.04	**0.23**
Elation	11.0	(3.6)	14.1	(4.3)	12.6	(4.0)	13.1	(4.1)	0.420	**0.006**	0.053	0.03	**0.30**	0.16
Calmness	11.6	(3.7)	15.5	(3.9)	12.9	(4.3)	14.5	(4.2)	0.767	**<0.001**	0.075	0.00	**0.51**	0.14
Fatigue	15.3	(5.5)	11.7	(4.7)	13.7	(5.6)	13.0	(4.9)	0.833	**0.004**	**0.035**	0.00	**0.32**	**0.19**
Depression	15.1	(5.3)	11.2	(4.9)	12.8	(5.1)	10.3	(4.0)	**0.004**	**<0.001**	0.223	**0.32**	**0.56**	0.07
Contemplation	13.8	(4.3)	12.5	(4.0)	13.7	(2.6)	12.2	(3.8)	0.759	**0.036**	0.765	0.00	**0.18**	0.00
Anger	12.3	(5.1)	8.1	(4.4)	10.0	(4.8)	8.3	(4.1)	**0.037**	**<0.001**	**0.030**	**0.18**	**0.62**	**0.20**
Excitement	14.5	(4.9)	10.1	(4.3)	12.2	(5.0)	9.6	(4.6)	**0.043**	**<0.001**	**0.028**	**0.17**	**0.46**	**0.20**

^a^: η²p: effect size partial η squared. Bold values indicate significant *p*-values/effect sizes.

## Data Availability

All relevant data supporting the conclusions of this article are included in the manuscript.

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
