# Peer review of "Acute Effects of a Single Bout of Walking on Affective Responses in Patients with Major Depressive Disorder"

_ijerph, 2021, doi:10.3390/ijerph18041524_

Round 1

Reviewer 1 Report

The authors tackled a research question highly relevant to our field, the work has marit. And I like the application of different mood/affect/emotion ratings. However, the quality of the manuscript can be improved.

Major points I suggest to consider:

a) The authors did not state any hypotheses, one would expect these to be presented in an unambigious manner at the end of the introduction.

b) The introduction is wordy and the key message can be reconsidered: Is there really a lack of studies or rather a lack in treatment approaches using exercise in psychiatry? 

b) The power of the data used to investigate the reserach question is at least a major limitation and should thus be stated clearly and straightforwardly.

c) The discussion section appears very hard to read to me. In other words, the structure is unusual. I would suggest to follow common procedures for reporting original reserach, i.e., first stating the essence of the own result, then naming and citing other reserach regarding this question (regardless if supportive or not), afterwards discussing why the results are (not) in line with the own ones, and lastly giving a broader context what this implies for further research. I just to follow this strcuture for each piece of the results step by step to enable a clear understanding for readers that are not that familiar with the field.

d) Models should be controlled for medication status of participants

Minor points:

a) Psychometric Properties of the scales used in the study should be reported 

b) Figure 2 is described insufficiently: What do dots exactly represent, which average?

c) please be straightforward in defining exercise - is it an exercise intervention you applied or rather a daily life activity?

d) in the abstract, please provide more statistical information, e.g., in the sentence starting w "Negatively valenced..."

e) the manuscript would benefit from native proofreading

Author Response

Thank you for your comments; please see the attachment. 

Reviewer 2 Report

  1. What is the innovation of this study compared with previous studies and what are the unique contributions of this study?
  2. The overall sample size is small, and this paper seems not to introduce the number of subjects in the experimental group and the control group, as well as the differences in demographic characteristics between the two groups
  3. In this study, the data of the scale was measured every 5 minutes during the process of the experiment. Such frequent interruption of the experimental process would not cause obstacles to the fluency of the experimental procedure, thus affecting the experimental results?
  4. The study does not appear to describe how to control the experimental environment and conditions, such as whether the subjects had taken antidepressant and anti-anxiety drugs before the study, or had other forms of therapeutic activity (e.g. psychological interventions). Was the experiment designed to exclude as many other factors as possible that might affect the results of the experiment?

Author Response

(The authors gave the same response as above.)

Reviewer 3 Report

Thank you for the opportunity to read this interesting manuscript. I think the study has the potential to better understand the relation between exercise and mental health. However, while reading the manuscript I came across several issues that need to be addressed before I believe the manuscript is suitable for publication. Please see below my suggestions to improve the manuscript.

  • Introduction
    • I notice that a lot of emphasis is placed on results of treatments of depression (e.g., antidepressants, cognitive behavioral therapy), while you investigate the acute effects of exercise (and not the effects of exercise treatments). Although you explain why studying acute effects is relevant in the context of current treatments, I think you can better balance what is known about acute effects of exercise and what is known about treatments of depression. I think more emphasis should be placed on what your study adds to knowledge about acute effects of exercise. That is, I am aware of other studies investigating the acute effects of exercise on affect. These earlier studies are not included in your introduction. How does your study relate to these earlier studies? What does your study add to these earlier studies?
    • The Circumplex model is explained shortly. Personally, I am familiar with the Circumplex Model, but I can imagine that not every reader is familiar with this model and may experience difficulties understanding the model. Maybe you can add some examples of dimension-specific affective states?
    • Line 101: what do you mean by ‘potential longer-term effects’? Hours, weeks, months?
    • Line 105: What is a ‘multi-approach’ method?
  • Materials and methods/Results
    • Line 126: the sentence “The procedure was identical in both conditions except for exercise/passive control” is not clear. Do you mean that the procedure was identical in both conditions, except for the activity?
    • Procedure: although it is visible in the Figure, maybe good to indicate in the text that the duration of the activity in each condition is 30 minutes.
    • Line 130: “compare” should be “see”.
    • Line 130: add “section” after “Measurements”
    • Not clear for the reader what “rotated dimensional affective responses” and “categorical affective responses” are.
    • Maybe it is an idea to first explain the conditions, and – after this – the procedure? This would increase understanding of the procedure-section.
    • Why did you choose for a self-selected light-moderate intensity? Is it truly a self-selected intensity, as you instructed participants to walk as if they were in a hurry to catch a bus? Why did you give this instruction?
    • Line 151-156: again, the dimensional and categorical approach of affect is not clear. Maybe add an example? (I see that you explain this in Line 166-169, maybe add this earlier in the methods section?)
    • Why did you choose for several measures of affect, and why are some affect measures not collected at all time points? Possibly, is this information that should be added to the introduction section?
  • Discussion
    • See my previous comments about dimensional and categorical affective responses/multimethod approach. This information is difficult to understand for readers who are not familiar with the (measurements of the) Circumplex model.
    • I would prefer more ‘normal language’ and less statistical language in the discussion section, i.e., what do the results mean/what did you learn from your study?
    • Line 270-273: here you state that previous studies in the clinical context already have shown that single bouts of moderate intensity exercise can increase affect. How does your study differ from these earlier studies? I.e., what does your study add to these previous studies? Why is this information not included in the introduction section?
    • Line 286-301: Here I read how your study extends existing knowledge. I think this information should be presented in the introduction, and, again discussed in the discussion section. In this way, the introduction and the discussion form a cohesive narrative, and it is clearer (in the beginning of your manuscript) what your study adds to the current body of knowledge.
    • Line 295-297: why does one measurement of affect not provide a comprehensive insight in affective responses, i.e., why do you need more measures?
    • Line 302-Line 305: this information could be included in the methods section as a rationale for the reason why you chose a self-selected intensity of physical activity.
    • Line 312-313: “The present findings…in accordance”. I doubt whether this correct English.
    • Line 312-321: I would also prefer a less statistical discussion about the (non-significant) findings. E.g., is it possible that people who sat quietly, had more cognitive capacity for rumination, and this could be related to increases in activation
    • Line 322-333: This is a little bit difficult to understand.
    • Line 347-349: You can still do a power analysis (although not a priori, but post-hoc), so I would not mention this point.
    • Line 349-355: Do you have ideas why the large numbers of completions of the Feeling and Felt Arousal Scales should be critically discussed? What are potential pitfalls?
    • Is selectivity bias a potential limitation? i.e., those who expressed interest in the study were included, i.e., maybe motivated to exercise.
    • Personally, I would combine limitations with suggestion for future research. How can future studies deal with the issues you list?

Author Response

(The authors gave the same response as above.)

Round 2

Reviewer 2 Report

After reviewing the manuscript again, I consider that the authors have done a great job, responding satisfactorily to all the questions requested.My advice is to accept.

Author Response

Thank you for your time and effort.

Reviewer 3 Report

I've read the revised manuscript and I am satisfied how the authors handled my suggestions, and I have no further suggestions. 

Author Response

Thank you for your time and effort.